# Investigating Links between Moderate-to-Vigorous Physical Activity and Self-Rated Health Status in Adolescents: The Mediating Roles of Emotional Intelligence and Psychosocial Stress

**DOI:** 10.3390/children10071106

**Published:** 2023-06-25

**Authors:** Huilin Wang, Yang Liu, Songbiao Zhang, Ziqing Xu, Jingyu Yang

**Affiliations:** 1School of Business, Hunan University of Science and Technology, Xiangtan 411201, China; 2International College, National Institute of Development Administration, Bangkok 10240, Thailand; 3Department of Medical Bioinformatics, University of Göttingen, 37077 Göttingen, Germany; jingyu.yang@bioinf.med.uni-goettingen.de

**Keywords:** moderate-to-vigorous physical activity, emotional intelligence, psychosocial stress, self-rated health status, adolescents

## Abstract

Adolescence represents a crucial phase, characterized by rapid physical and mental development and numerous challenges. Physical activity plays a vital role in the mental well-being of adolescents; however, due to the prevailing educational philosophy prioritizing academic performance, adolescent participation in physical activities has yet to reach its full potential. Thus, this study aims to investigate the effects of moderate-to-vigorous physical activity on adolescents’ emotional intelligence, psychosocial stress, and self-rated health status. To achieve this objective, a cluster sampling method was employed to collect data from 600 adolescents in 10 schools across five municipal districts of Changsha, China. A total of 426 valid questionnaires were returned and analyzed. Utilizing AMOS v.23, a structural equation model was constructed to validate the hypotheses. The findings reveal that moderate-to-vigorous physical activity significantly impacts adolescents’ emotional intelligence and self-rated health status. Conversely, it exerts a significant negative influence on their psychosocial stress. Moreover, emotional intelligence and psychosocial stress mediate the relationship between moderate-to-vigorous physical activity and self-rated health status. In light of these results, education departments, schools, and families must embrace a paradigm shift in educational philosophies and provide robust support for adolescents to engage in moderate-to-vigorous physical activities.

## 1. Introduction

In recent years, the World Health Organization (WHO) and the United Nations (UN) have expressed growing concern regarding adolescents’ physical and mental well-being. Adolescence is a crucial developmental stage, and poor health can have long-lasting effects [1]. This phase is characterized by a range of physical, emotional, and psychological changes significantly influencing an individual’s personality and behavior, as it signifies the transition from childhood to adulthood [2]. According to the WHO, half of all mental health disorders originate by age 14, with three-quarters emerging by age 24 [3]. Additionally, rates of physical inactivity, obesity, and substance abuse tend to escalate during adolescence. Disturbingly, a substantial number of adolescents fail to engage in sufficient physical activity, leading to health problems associated with obesity [4].

Furthermore, mental health issues such as depression and anxiety are prevalent among this age group [5]. The increasing rates of adolescent depression, sub-health, and mental illness have become a growing concern for many healthcare professionals and policymakers [6]. This may be attributed to adolescents’ exposure to a wide range of stressors, including academic pressure, social media, and family conflict [7]. These stressors can contribute to anxiety, depression, and other mental health problems [8]. Moreover, the prevalence of digital devices has resulted in increased sedentary behavior and reduced physical activity, which can further contribute to obesity and other health issues [9]. The combination of these factors poses significant challenges to the overall well-being of adolescents.

Chinese adolescents’ lack of physical activity can be attributed to several direct factors, including a heavy schoolwork load, intense pressure to enter higher schools, inadequate sports facilities, and a lack of a sports-oriented atmosphere [10,11]. Unfortunately, both schools and parents tend to prioritize academic performance over sports activities, resulting in a decline in the physical fitness of adolescents [12], an increase in obesity rates [13], and even vision problems [14]. This negative trend continues to persist, with the proportion of overweight students rapidly rising. In urban areas, nearly a quarter of boys are classified as overweight or obese, and more than two-thirds of students suffer from poor eyesight [15]. If this situation remains unaddressed, it will undoubtedly have detrimental effects on the overall health of adolescents.

Physical activity plays a crucial role in the physical, mental, and emotional development of adolescents [16]. It not only promotes better sleep but also improves cognitive function and boosts self-esteem [17]. Adolescents who actively engage in physical activity are more likely to develop lifelong healthy habits [18]. Interestingly, recent evidence suggests that moderate-to-vigorous physical activity (MVPA) provides greater benefits for adolescents compared to normal physical activity [19]. MVPA encompasses activities that raise heart rate and breathing rates, such as running, swimming, or cycling [20]. Numerous studies have demonstrated the multiple advantages of MVPA for adolescents, including improved physical health, enhanced cognitive function, and better academic performance [21]. Additionally, MVPA has been found to improve cardiovascular health, increase muscle and bone strength, and to facilitate healthy weight management [22]. While ample research supports the advantages of MVPA, further investigation is required to understand precisely how MVPA exerts positive effects on the physical and mental well-being of adolescents.

Psychosocial stress is a common experience for adolescents as they navigate academic pressures, social challenges, and family conflicts [23]. The long-term effects of chronic stress can be detrimental to both physical and mental health, increasing the risk of chronic diseases and mental health disorders [24]. MVPA has shown promising results in reducing adolescent stress levels by promoting relaxation, improving sleep quality, and fostering overall well-being [25,26]. Additionally, adolescence is a critical period for developing emotional intelligence, which involves recognizing and regulating one’s own emotions, as well as understanding the emotions of others [27]. During this stage, individuals navigate complex social relationships and experience a wide range of emotions [28]. Research suggests that regular participation in MVPA may enhance emotional intelligence in adolescents, leading to improvements in mood, reduced anxiety and depression, increased self-esteem, stronger interpersonal relationships, better academic performance, and a reduced risk of mental health issues [29]. Previous studies have indicated that emotional intelligence and psychosocial stress act as mediators in the relationship between MVPA and self-rated health status [30,31].

Previous studies have primarily focused on the impact of physical activity on the physical health of adolescents [32,33]. However, this research shifts attention towards examining adolescents’ physical and mental health aspects, with self-rated health status as the dependent variable. Self-rated health status encompasses the evaluation of both physical and mental well-being [34]. It is a subjective measure reflecting an individual’s overall perception of their health and well-being [35]. Various factors, including levels of physical activity influence this measure. Research has shown that adolescents who engage in regular MVPA tend to report higher levels of self-rated health status compared to those who are less active [36]. However, despite the potential benefits of MVPA for emotional intelligence, psychosocial stress, and self-rated health status in adolescents, there are still gaps in our understanding of how these factors are interrelated. The objective of this research is to explore the underlying mechanisms that contribute to these relationships and to identify strategies for promoting MVPA among adolescents.

Therefore, this research proposes the following hypotheses:

**Hypothesis** **1** **(H1).**
*Moderate-to-vigorous physical activity is associated with adolescents’ emotional intelligence.*


**Hypothesis** **2** **(H2).**
*Moderate-to-vigorous physical activity is associated with adolescents*
*’ psychosocial stress.*


**Hypothesis** **3** **(H3).**
*Moderate-to-vigorous physical activity is associated with adolescents*
*’ self-rated health status.*


**Hypothesis** **4** **(H4).**
*Adolescents*
*’ emotional intelligence is associated with psychosocial stress.*


**Hypothesis** **5** **(H5).**
*Adolescents*
*’ emotional intelligence is associated with their self-rated health status.*


**Hypothesis** **6** **(H6).**
*Adolescents*
*’ psychosocial stress is associated with their self-rated health status.*


**Hypothesis** **7** **(H7).**
*Emotional intelligence and psychosocial stress mediate the relationship between moderate-to-vigorous physical activity and self-rated health status.*


A summary of all hypotheses is shown in Figure 1.

## 2. Materials and Methods

### 2.1. Participants and Procedure

The study employed the cluster sampling method. Paper questionnaires were distributed in one junior high school and one high school in each of the five municipal districts of Changsha City, China from February to March 2023. Within each grade, participants were randomly selected from various classes. It is important to note that all students participated in the survey voluntarily, and signed informed consent was obtained from their parents prior to their participation. As a token of appreciation, all participating students received a stationery item upon completing the questionnaire. A total of 600 adolescents from 10 schools were selected as the survey participants, yielding 426 valid questionnaires. The recovery rate for valid questionnaires was 85.2%.

### 2.2. Measurement

#### 2.2.1. Moderate-to-Vigorous Physical Activity (MVPA)

MVPA was assessed using the Physical Activity Scale developed by Sallis et al. [37] and Andersen et al. [38]. The scale consisted of four items measured on a 5-point Likert scale, ranging from “strongly disagree (1)” to “strongly agree (5)”. The scale questions were structured as declarative sentences and employed a first-person inquiry method. For instance, the items included statements such as “During the past week, I actively participated in various forms of moderate physical activity, including tasks like sweeping and mopping, as well as engaging in sports such as volleyball, Ping-Pong, and similar activities”.

#### 2.2.2. Emotional Intelligence

The emotional intelligence of adolescents was evaluated using the Emotional Intelligence Scale developed by Law et al. [39]. This inventory comprises three items, each rated on a 5-point Likert scale ranging from “strongly disagree (1)” to “strongly agree (5)”. The scale questions were constructed as declarative sentences, adopting a first-person inquiry method. For instance, an example item is “I am capable of regulating my emotions effectively to address challenges in a rational manner”.

#### 2.2.3. Psychosocial Stress

Psychosocial stress levels among adolescents were evaluated using a subset of three items from the Global School-Based Student Health Survey, developed by the World Health Organization (WHO) [40]. These three items focused on assessing the negative stress perception, feelings of loneliness, and trouble sleeping. A 5-point Likert scale was utilized, ranging from “never (1)” to “always (5)” to gauge the frequency of experiences. The scale questions were structured as interrogative sentences, adopting a second-person inquiry method. For instance, an example item would be “During the past week, how frequently did you experience worries that prevented you from sleeping at night?”

#### 2.2.4. Self-Rated Health Status

The researchers extended the scope of the project initiated by the Ontario Student Drug Use Survey (OSDUS) to evaluate the self-rated health status of adolescents [41]. In addition to physical health, mental health was incorporated into the assessment. The scale comprises five items, employing a 5-point Likert scale ranging from “poor (1)” to “very good (5)” to gauge the perceived status. The scale questions are formulated as interrogative sentences, utilizing a second-person questioning method. For instance, a sample item reads as follows “How would you rate your mental health?”

### 2.3. Data Analysis

Using AMOS v.23, this study developed a structural equation model (SEM) to examine the impact of MVPA on the self-rated health status of adolescents. The maximum likelihood (ML) estimation method was employed to estimate the model parameters. A two-step modeling approach, as suggested by Anderson and Gerbing [42], was utilized to evaluate both the measurement and structural models. The initial step involved a comprehensive assessment of the model’s reliability and validity. This involved evaluating the measurement model and ensuring the reliability of the latent constructs. The subsequent step encompassed measuring the fit indices and path coefficients of the hypothesized model to examine the presence of a mediating effect.

To address the concern of common method variance (CMV) inherent in self-reported behaviors, the researchers followed the suggestion of Mossholder et al. [43]. This study conducted a comparison between model one and model two, examining the differences in degrees of freedom and chi-square values. The results indicated that the chi-square value for model one was 2455.645, with 90 degrees of freedom and a *p*-value less than 0.001. Similarly, the chi-square value for model two was 218.682, with 84 degrees of freedom and a *p*-value less than 0.001. These findings indicate that model one fits proportionally to model two. Therefore, it can be concluded that there was no evidence of univariate structure, suggesting that CMV was not a concern in this study.

## 3. Results

### 3.1. Assessment of the Measurement Model Reliability and Validity

To assess the reliability and discriminant validity of the latent variables, Cronbach’s alpha and composite reliability (CR) coefficients were calculated in this study [44]. The obtained results are presented in Table 1. The Cronbach’s alpha coefficients for all four scales ranged from 0.919 to 0.961, indicating high internal consistency. Furthermore, the CR values for all variables exceeded the threshold of 0.9, affirming excellent reliability. Additionally, the average variance extracted (AVE) values for all variables fell within the range of 0.798 to 0.861, demonstrating strong convergent validity. Moreover, the correlation coefficients, as shown in Table 2, were observed to be lower than the square root of AVE for each variable. This finding indicates good discriminant validity among all variables, suggesting that they measure distinct constructs. Overall, the results from the analysis confirm that all variables exhibit high reliability, convergent validity, and satisfactory discriminant validity.

### 3.2. Hypothesis Testing Results

Firstly, the error term and residual term of the structural equation model were not observed to have negative values, indicating that the estimation of the model was not violated. Secondly, the fit of the data to the structural equation model was deemed high, as indicated by various fit indices, χ^2^/df = 2.603, GFI = 0.933, AGFI = 0.904, NFI = 0.970, CFI = 0.981, TLI = 0.977, and RMSEA = 0.061. Thirdly, the Pearson correlation results presented in Table 2 demonstrated a significant correlation among the independent variable, mediator variable, and dependent variable, thereby supporting the hypothesis verification. Fourthly, the structural path model is visually represented in Figure 2. The impact of MVPA on emotional intelligence was found to be statistically significant (*β* = 0.600, *p* < 0.001), thereby providing support for H1. Additionally, the effect of MVPA on psychosocial stress was statistically significant (*β* = −0.404, *p* < 0.001), supporting H2. Furthermore, MVPA was observed to have a statistically significant effect on self-rated health status (*β* = 0.347, *p* < 0.001), thereby supporting H3. The effect of emotional intelligence on psychosocial stress was also statistically significant (*β* = −0.395, *p* < 0.01), supporting H4. Likewise, the effect of emotional intelligence on self-rated health status was statistically significant (*β* = 0.332, *p* < 0.001), supporting H5. Finally, the effect of psychosocial stress on self-rated health status was observed to be statistically significant (*β* = −0.265, *p* < 0.001), thereby supporting H6.

The researchers posited a hypothesis suggesting that the influence of MVPA on self-rated health status is indirectly mediated by two factors, emotional intelligence and psychosocial stress. In this study, the researchers employed the Bootstrap method to examine the existence of a mediating effect [45]. The results, presented in Table 3, display the standardized outcomes of the 95% confidence interval from the 5000-bootstrap sample. The findings revealed that both emotional intelligence and psychosocial stress significantly contribute to the relationship between MVPA and self-rated health status (standardized indirect effect = 0.369, *p* < 0.001), thereby providing support for H7. These results suggest that adolescents who engage in more MVPA, exhibit higher levels of emotional intelligence, and experience lower levels of psychosocial stress, tend to report better self-rated health status.

## 4. Discussion

### 4.1. Theoretical Contribution

This study makes significant contributions to the theoretical analysis of adolescent physical activity. Firstly, while previous research has predominantly focused on the effects of physical activity on adolescent health [46] and academic performance [47], this study addresses a critical research gap by examining the impact of MVPA on adolescents’ emotional intelligence, psychosocial stress, and self-rated health status. Adolescence is a period characterized by rapid physical development, sexual and biological maturity, and an intensified sense of self. Consequently, adolescents face a myriad of pressures resulting from their fast-paced physical changes and relatively immature psychology. Moreover, they contend with academic pressures, high parental expectations, and intricate peer relationships, all of which impact their physical and mental well-being to varying degrees. Engaging in MVPA has been found to have a profoundly positive effect on the physical and mental health of adolescents. Participating in sports not only helps them alleviate stress, but also facilitates the establishment of positive interpersonal relationships. Additionally, it enhances their emotional intelligence and equips them with better control over negative emotions.

Secondly, the findings of this study provide compelling evidence for the profound influence of MVPA on emotional intelligence, psychosocial stress, and self-rated health status among adolescents. Notably, MVPA exerted the most substantial influence on emotional intelligence, followed by psychosocial stress, while its impact on self-rated health status was comparatively lower. This observation can be attributed to the mediating role played by emotional intelligence and psychosocial stress in the relationship between MVPA and self-rated health status. MVPA has a direct impact on adolescents’ perceived health status, but it can also indirectly influence their perceived health status by affecting emotional intelligence and psychosocial stress levels. Figure 2 visually illustrates the relationships among these variables, revealing that emotional intelligence and psychosocial stress collectively accounted for an impressive 66.8% of the variance in self-rated health status. This highlights the significant role played by these factors in shaping adolescents’ overall perception of their own health.

In summary, this study fills a crucial research gap by expanding the focus beyond the traditional domains of adolescent physical activity. By exploring the effects of MVPA on emotional intelligence, psychosocial stress, and self-rated health status, it sheds light on the comprehensive impact of MVPA on adolescents’ overall well-being.

### 4.2. Practical Implications

Given the significant impact of MVPA on adolescents’ emotional intelligence, social and interpersonal relationships, as well as self-rated health status, it is essential for the government, schools, and families to provide comprehensive support for adolescents’ engagement in physical activities.

First and foremost, the education department should recognize the pivotal role of sports in promoting the physical and mental well-being of adolescents. Efforts should be made to enhance the prominence of sports within the curriculum, such as incorporating sports-related content into the college entrance examination and increasing the weightage of sports scores in the high school entrance examination. By doing so, the education system can effectively underscore the importance of physical activity and encourage its inclusion as a core aspect of academic evaluation. Furthermore, it is imperative to reinforce the supervision of physical education in schools and implement various measures that foster the development of physical activity among adolescents. This includes ensuring that physical education programs are adequately structured and effectively implemented. Moreover, initiatives should be undertaken to enhance awareness of the broader society regarding the benefits of physical activity, thereby promoting a culture that values and supports the engagement of young individuals in regular physical exercise. Therefore, recognizing the profound impact of MVPA on adolescents’ overall well-being, it is crucial for the government, schools, and families to collaborate and provide comprehensive support for adolescents’ involvement in physical activities. This involves enhancing the status of sports within the curriculum, strengthening supervision of physical education in schools, and raising awareness within society about the importance of physical activity.

Secondly, it is crucial for schools to prioritize the reinstatement of physical education courses, ensuring that students are provided with sufficient exercise time during school hours. In China, many schools tend to place greater emphasis on students’ academic performance rather than fostering their holistic development, such as their overall abilities and potential for admission into esteemed high schools and universities. As a result, physical education is often overlooked, either substituted with other demanding subjects or lacking essential training and evaluation requirements for students. These issues demand serious attention from both the education department and schools, with a need to issue warnings and hold accountable those schools that perpetuate such problems. Furthermore, in addition to teaching students essential sports skills and scientific fitness knowledge, physical education teachers should also cultivate students’ interest and habits in sports. Many students express their dislike for sports due to feelings of inadequacy in certain activities, resulting in a lack of enjoyment and a sense of achievement. To address this, physical education teachers should tailor sports activities to suit the diverse characteristics of different students, catering to their individual strengths and preferences. Therefore, it is imperative for schools to prioritize physical education courses, ensuring that students receive the necessary exercise time during school hours. The education department and schools must address the tendency to prioritize academic performance over comprehensive development. Physical education teachers should not only impart sports skills and fitness knowledge, but also foster students’ interest and habits in sports by offering diverse activities that align with each student’s unique characteristics.

Thirdly, the enhancement of adolescents’ physical activity is closely intertwined with the support provided by families, making the concept of family physical education pivotal. In China, it is unfortunately common for many parents to overlook the importance of sports, believing that adolescents’ academic performance takes precedence and that investing time and resources in physical exercise is unnecessary. This misguided educational mindset is detrimental to the holistic development of adolescents’ physical and mental well-being. Researchers assert that rectifying this issue necessitates a shift in family education towards promoting a correct understanding of the significance of sports. An exemplary model in this regard is “Parents as Sports Instructors” advocated by the National Alliance for Youth Sports [48]. This initiative emphasizes that parents not only refrain from discouraging adolescents’ participation in physical activities but also actively guide and accompany them in such pursuits. This approach demonstrates the profound commitment of American society to fostering family involvement in sports, while significantly contributing to the physical and mental health of American adolescent. This endeavor serves as a valuable lesson and inspiration for many nations worldwide.

### 4.3. Limitations

This study has certain limitations that should be acknowledged. Firstly, it is important to note that this study employed a cross-sectional survey design, which restricts the ability to establish causality or determine temporal relationships. Additionally, the researchers focused on a limited number of factors that influence adolescents’ self-rated health status, and future research should aim to include a more comprehensive range of factors. It is recommended that future studies employ a mixed-methods approach, combining qualitative and quantitative research methods, to provide a more nuanced understanding of the topic. This would enable a deeper exploration of potential model variations and further development of the research framework.

Secondly, it is important to recognize that there may exist a disparity between self-rated health status and actual health status. In future research, it would be valuable to consider the real physical and mental health status of adolescents as a dependent variable. This would involve conducting assessments that provide more objective measurements of physical and mental health, allowing for a more robust examination of the impact of MVPA on adolescents’ overall well-being.

Thirdly, it is important to note that this study did not incorporate gender as a moderating variable. Future research should explore whether the impacts of MVPA on self-rated health status vary across different gender groups. Additionally, it may be valuable to investigate the effects of other demographic characteristics on these relationships. By considering a broader range of variables, we can gain a more comprehensive understanding of the factors influencing the association between MVPA and self-rated health status.

Fourthly, MVPA was assessed using the scales developed by Sallis et al. [37] and Andersen et al. [38]. It is worth noting that while accelerometer are commonly used in academia for measuring MVPA [49,50], this study did not incorporate their use. Therefore, future research should consider employing accelerometer as a more scientifically and accurate method for measuring MVPA. The utilization of accelerometers would provide a more objective and precise assessment of individuals’ physical activity levels.

In summary, while this study provides valuable insights, it is important to acknowledge its limitations. Future research should adopt longitudinal designs, consider a wider range of influencing factors, incorporate qualitative research methods, and incorporate more objective measures of physical and mental health status to further advance our understanding of the relationship between MVPA and adolescents’ well-being.

## 5. Conclusions

The findings of this study highlight the significant impact of MVPA on adolescents’ emotional intelligence, psychosocial stress, and self-rated health status. As a result, education departments, schools, and families must prioritize adolescents’ physical and mental well-being, reassess educational paradigms, and provide necessary resources, facilities, and time support to promote physical activities among this age group.

Based on these findings, this study strongly recommends that education departments actively acknowledge and address adolescents’ physical and mental health concerns. This entails reevaluating existing educational concepts and policies to ensure they prioritize holistic well-being alongside academic achievements. Additionally, it is essential for schools to allocate adequate funding, provide suitable venues, and dedicate sufficient time within the curriculum to facilitate regular physical activity for students.

Moreover, families play a vital role in supporting adolescents’ physical activities. Parents and caregivers must recognize the importance of physical exercise for their children’s overall development and well-being. They should encourage and actively participate in physical activities with their adolescents, fostering a supportive and engaging environment at home.

In conclusion, the findings of this study confirm the positive impact of MVPA on adolescents’ emotional intelligence, as well as its negative impact on psychosocial stress. Furthermore, MVPA has been shown to have a positive effect on adolescents’ self-rated health status. These results emphasize the importance of prioritizing the physical and mental well-being of young individuals. It is imperative for education departments, schools, and families to shift their educational paradigms, allocate resources accordingly, and unwaveringly support young people’s engagement in sports activities. By doing so, we can promote the holistic health of the younger generation.

## Figures and Tables

**Figure 1 children-10-01106-f001:**
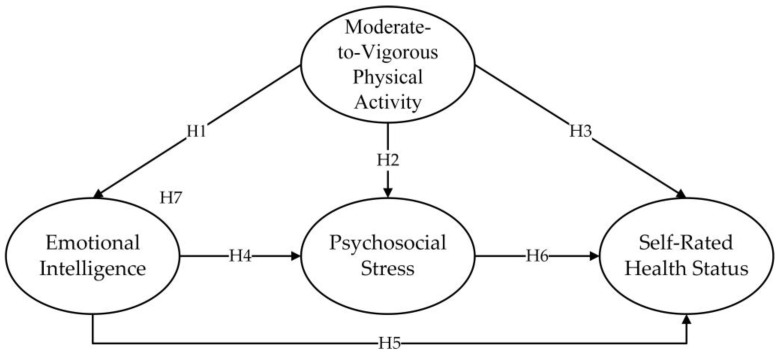
The hypothesized model.

**Figure 2 children-10-01106-f002:**
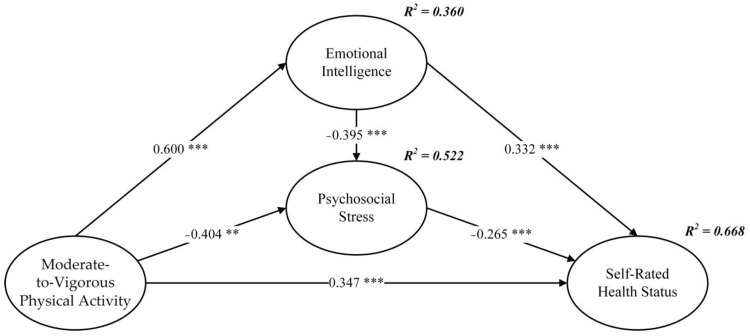
Structural path model. ** *p* < 0.01, *** *p* < 0.001. Standardized coefficients are reported.

**Table 1 children-10-01106-t001:** Reliability and validity test.

Items	Loadings	Cα	AVE	CR
** *Moderate-to-Vigorous Physical Activity (MVPA)* **		0.961	0.861	0.961
MVPA1	0.933			
MVPA2	0.921			
MVPA3	0.935			
MVPA4	0.922			
** *Emotional Intelligence (EI)* **		0.919	0.798	0.922
EI1	0.884			
EI2	0.905			
EI3	0.891			
** *Psychosocial Stress (PS)* **		0.933	0.825	0.934
PS1	0.920			
PS2	0.885			
PS3	0.919			
** *Self-Rated Health Status (SRHS)* **		0.957	0.819	0.958
SRHS1	0.922			
SRHS2	0.914			
SRHS3	0.924			
SRHS4	0.866			
SRHS5	0.898			

**Table 2 children-10-01106-t002:** Discriminant validity test.

Construct	MVPA	EI	PS	SRHS
MVPA	**(0.928)**			
EI	0.567 **	**(0.893)**		
PS	−0.605 **	−0.591 **	**(0.908)**	
SRHS	0.683 **	0.667 **	−0.656 **	**(0.905)**

The square root of the average various extracted (AVE) is in diagonals (bold); off diagonals are a Person’s corrections of contracts. ** *p* < 0.01.

**Table 3 children-10-01106-t003:** Standardized indirect effects.

	Point Estimate	Product of Coefficients		Bootstrapping		
				Bias-Corrected 95% CI		Two-Tailed Significance
		*SE*	*Z*	Lower	Upper	
MVPA → SRHS	0.369	0.037	9.973	0.299	0.445	0.000 (***)

Standardized estimations of 5000 bootstrap samples. *** *p* < 0.001.

## Data Availability

The data that support findings and conclusions of this study will be available from the corresponding author upon a reasonable request.

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
