# Peer review of "Investigating Links between Moderate-to-Vigorous Physical Activity and Self-Rated Health Status in Adolescents: The Mediating Roles of Emotional Intelligence and Psychosocial Stress"

_children, 2023, doi:10.3390/children10071106_

Round 1

Reviewer 1 Report

This is a well-written manuscript using a large sample size of Chinese adolescents to assess the association between MVPA and self-rated health status. The sample was selected through the cluster sampling method and statistics seem to be conducted correctly. In general, the article is of interest to a wide audience, especially researchers in the field of mental health and physical activity.

My suggestions for improving the manuscript mainly concern some methodological aspects and limitations to be clarified/improved: 

1-     Despite the importance of gender in physical activity practice and mental health, the analysis did not consider this factor in any way. How many were the girls? And the boys? Why not check whether different patterns were present between them?  

2-     It is well known that reliable assessments of MVPA are only obtained with the use of accelerometers. Taking into account the literature (e.g. look at doi.org/10.3390/su13010335  or  doi: 10.1371/journal.pone.0179429 )  discuss this issue and report it as one of the limitations of the study.

None

Author Response

Dear reviewer,

Thank you for your thorough review and valuable feedback on our manuscript entitled "Investigating links between moderate-to-vigorous physical activity and self-rated health status in adolescents: The mediating roles of emotional intelligence and psychosocial stress". We appreciate the time and effort you have dedicated to providing insightful comments that have undoubtedly improved the quality of our research. In response to your suggestions, we have carefully considered and made revisions accordingly. We would like to address each of your points and explain the changes we have implemented.

  1. Despite the importance of gender in physical activity practice and mental health, the analysis did not consider this factor in any way. How many were the girls? And the boys? Why not check whether different patterns were present between them?

Response: Thank you for your suggestion. Indeed, we acknowledge that the omission of gender-related analysis is a notable limitation of our research design. As you rightfully pointed out, the questionnaire used in our study did not include a specific inquiry about the gender of the participating adolescents. This limitation restricts our ability to explore potential variations in the impact of MVPA on self-rated health status across different gender groups.

In response to your suggestion, we have included a discussion of this limitation in the "4.3 Limitations" section of the manuscript. The added content reads as follows: "Thirdly, it is important to note that this study did not incorporate gender as a moderating variable. Future research should explore whether the impacts of MVPA on self-rated health status vary across different gender groups. Additionally, it may be valuable to investigate the effects of other demographic characteristics on these relationships. By considering a broader range of variables, we can gain a more comprehensive understanding of the factors influencing the association between MVPA and self-rated health status."

  1. It is well known that reliable assessments of MVPA are only obtained with the use of accelerometers. Taking into account the literature (e.g. look at doi.org/10.3390/su13010335 or doi: 10.1371/journal.pone.0179429 )  discuss this issue and report it as one of the limitations of the study.

Response: Thank you for your suggestion. We agree with your observation that the use of accelerometers is a more commonly adopted method in academia for measuring MVPA due to its objectivity and precision. Regrettably, our study did not incorporate the use of accelerometers, which is indeed a limitation that hampers the accuracy of our MVPA assessment.

To address this limitation, we have included a discussion of this point in the "4.3 Limitations" section of the manuscript. The added content reads as follows: "Fourthly, MVPA was assessed using the scales developed by Sallis et al. [37] and Andersen et al. [38]. It is worth noting that while accelerometers are commonly used in academia for measuring MVPA [49,50], this study did not incorporate their use. Therefore, future research should consider employing accelerometers as a more scientifically and accurate method for measuring MVPA. The utilization of accelerometers would provide a more objective and precise assessment of individuals' physical activity levels."

Once again, we express our sincere gratitude to you, for taking the time to provide us with your valuable suggestions and feedback. We are truly grateful for your contribution to our research.

We would also like to extend our best wishes to you. We appreciate your dedication and commitment to the peer review process, which plays a vital role in ensuring the rigor and integrity of scientific publications.

Reviewer 2 Report

This study describes the effects of moderate-to-vigorous physical activity and self-rated health status in adolescents. While this topic is relevant is to investigate I have some concerns:

1. The conclusion "the study findings underscore the signfiicance of MVPA in shaping adolescents'emotional intelligence, psychosocial stress, and self-=assessed health status..." has both negative and positive valence aspects in it. Please redefine.

2. the number of participants is discrepant between abstract and methods

3. the hypothesis are formulated two-sided, while there is a clear conceptual idea. Please rephrase. 

4. Explained variances are quite divergent (.3 - .6). Please stress this more: the link between activity and intelligence is much higher than the link between activity and health status. 

quite ok. Minor issues

Author Response

Dear reviewer,

Thank you for your thorough review and valuable feedback on our manuscript entitled "Investigating links between moderate-to-vigorous physical activity and self-rated health status in adolescents: The mediating roles of emotional intelligence and psychosocial stress". We appreciate the time and effort you have dedicated to providing insightful comments that have undoubtedly improved the quality of our research. In response to your suggestions, we have carefully considered and made revisions accordingly. We would like to address each of your points and explain the changes we have implemented.

  1. The conclusion "the study findings underscore the signfiicance of MVPA in shaping adolescents' emotional intelligence, psychosocial stress, and self-assessed health status..." has both negative and positive valence aspects in it. Please redefine.

Response: Thank you for your suggestion. As per your recommendation, we have revised the final paragraph of the conclusion to accurately reflect the study's findings. The revised paragraph now reads as follows: "In conclusion, the findings of this study confirm the positive impact of MVPA on adolescents' emotional intelligence, as well as its negative impact on psychosocial stress. Furthermore, MVPA has been shown to have a positive effect on adolescents' self-rated health status. These results emphasize the importance of prioritizing the physical and mental well-being of young individuals. It is imperative for education departments, schools, and families to shift their educational paradigms, allocate resources accordingly, and unwaveringly support young people's engagement in sports activities. By doing so, we can promote the holistic health of the younger generation."

  1. the number of participants is discrepant between abstract and methods.

Response: Thank you for your valuable feedback. We appreciate your insights regarding the inaccuracies in our manuscript. As suggested, we have made the necessary revisions in the corresponding section of the abstract. The revised statement now accurately reflects our methodology: "To achieve this objective, a cluster sampling method was employed to collect data from 600 adolescents in 10 schools across 5 municipal districts of Changsha, China. A total of 426 valid questionnaires were returned and analyzed."

  1. the hypothesis are formulated two-sided, while there is a clear conceptual idea. Please rephrase.

Response: Thank you for your suggestion. We have made the necessary modifications as per your recommendation.

  1. Explained variances are quite divergent (.3 - .6). Please stress this more: the link between activity and intelligence is much higher than the link between activity and health status.

Response: Thank you for your valuable feedback on our manuscript. We appreciate your suggestion and have taken it into consideration. In Section 4.1, we have provided an explanation for why MVPA has a greater impact on emotional intelligence compared to its impact on self-rated health status. Specifically, we stated: "Notably, MVPA exerted the most substantial influence on emotional intelligence, followed by psychosocial stress, while its impact on self-rated health status was comparatively lower. This observation can be attributed to the mediating role played by emotional intelligence and psychosocial stress in the relationship between MVPA and self-rated health status. MVPA has a direct impact on adolescents' perceived health status, but it can also indirectly influence their perceived health status by affecting emotional intelligence and psychosocial stress levels."

Once again, we express our sincere gratitude to you, for taking the time to provide us with your valuable suggestions and feedback. We are truly grateful for your contribution to our research.

We would also like to extend our best wishes to you. We appreciate your dedication and commitment to the peer review process, which plays a vital role in ensuring the rigor and integrity of scientific publications.

Round 2

Reviewer 1 Report

The authors took my suggestions into account. The work is ready for publication. Congratulations!

None

Reviewer 2 Report

thank you

no issue